

# 1 Analysis of how dry-hot wind hazard has changed for winter

# 2 wheat in the Huang-huai-hai plain

Benlin Shi[a], Xinyu Zhu[a*], Hongzhong Li[a] , Yunchuan Hu[a], Yi Zhang[b]
[a] Shangqiu Normal University, Shangqiu 476000, Henan, China;
[b] Shangqiu Weather Bureau, Shangqiu 476000, Henan, China
**Corresponding author:** Xinyu Zhu
**Tel:** +86 370 2592902; **Email:** tia20021201@163.com
**Address:** College of Environment and Planning, Shangqiu Normal University, NO.298, Wenhua Road,
Shangqiu, 476000, Henan, China
**Abstract**: Climate change is exerting significant impacts on global agricultural production. Climatic
variations adversely affect crop production and, thus tend to impose a key constraint of agricultural
production, primarily on how to continuously enhance the winter wheat yields worldwide. The high
uncertainties in predicting the effects of climate change on wheat production are most likely due to
rare understanding on the responses of wheat production to extreme climatic factors, e.g. high
temperatures, low humidity as well as high wind speed. Dry-hot wind hazard represents one of the
main natural disasters for Chinese winter wheat production, especially for the Huang-huai-hai plain.
However, high uncertainties of the effects of dry-hot wind hazard on winter wheat production still
exist, mainly due to the gaps of long-term observations. Therefore, we selected Shangqiu as the case
study area to determine the occurrence regularity of dry-hot wind hazard on winter wheat production
in Huang-huai-hai plain. We analyzed regional meteorological data with daily resolution in the later
growth stage of winter wheat during the period of 1963 to 2012. In accordance with the
meteorological industry standards of "Disaster Grade of Dry-hot Wind for Wheat" by the China



Meteorological Administration, we synthesized analyzed the distribution of annual average days of
dry-hot wind in winter wheat growing seasons and the associated responses to the climate change.
Hence the relationships between dry-hot wind times and winter wheat yields were also discussed. The
results showed that the annual average days of light and severe dry-hot wind exhibited tended to
decline in the recent 50 years. Great inter-annual variations of light and severe dry-hot wind were
observed. The significant inter-annual variations were related with the corresponding meteorological
conditions of temperature, moisture and wind speed. The most serious damages of light and severe
dry-hot wind both occurred in 1960s while the damages appeared less in the 1980s and the last decade,
which could be also explained by the corresponding temperature, moisture and wind speed conditions.
From 1963 to 2012, a climatic mutation point of daily maximum temperature was found near 1972,
but insignificantly ($p>0.05$). The wind speed at 2:00 pm and the relative humidity at 2:00 pm were
closely related to the hazard conspicuously. A climatic mutation point of the wind speed at 2:00 pm
was found near 1984, and climatic mutation of the relative humidity at 2:00 pm was found near 1981
($p<0.05$). Daily maximum temperature, wind speed at 2:00 pm and the relative humidity at 2:00 pm
played a major role in decreasing trend of dry-hot wind disaster, and the significantly decreased of
wind speed at 2:00 pm constituted a main factor in Shangqiu. Dry-hot wind hazard is very sensitive to
climate change. Yields of winter wheat were negatively correlated with annual average days of
dry-hot wind in Shangqiu ($p<0.05$). In actual practices, great concerns should be paid on the defense
of dry-hot wind for winter wheat production.   Thus the most effective practices have to be taken for
enhancing the resistance of winter wheat to dry-hot wind hazard through improving filed
microclimate condition.
**Keywords**: climate change; Shangqiu; winter wheat; grain filling stage; dry-hot wind



## 1. Introduction

Climate change has led to the frequent occurrence of extreme weather. It was noted in the 4th and 5th

evaluation report of the Intergovernmental Panel on Climate Change (IPCC) that global warming has

exerted widespread effects on agricultural ecosystems, brought increasing uncertainties to agricultural

production, led to more frequent regional occurrences of meteorologically caused agricultural

disasters, and altered the planting patterns of crops (Qin, 2009; Lobell et al., 2012; Ge et al., 2012;

IPCC, 2007; 2013). As one of the major meteorological disasters disrupting winter wheat growth and

yield, dry-hot wind frequently occurs during wheat's flowering and grouting stages, giving rise to a

10%-20% yield loss of winter wheat in the years when its disastrous effects are severe (Liu et al.,

2012).

In recent years, some papers were published in the worldwide authoritative journal Nature, analyzed

the effects of climatic changes and meteorological disasters on wheat, reporting that climatic warming

and extreme drought resulted in early maturation, yield loss, and decline in dry matter accumulation

of wheat (Lobell et al., 2012; Pongratz et al., 2012; Basso et al., 2014; Asseng et al., 2015). These

papers indicate that the effects of climatic changes on food crops have become an important subject of

global research. Since the foundation of the new China, Chinese agriculturalists and meteorologists

have acquired fruitful achievements in dry-hot wind research (Chen et al., 2001; Liu et al., 2008;

Wang et al., 2010; Liu et al., 2012; Zhao et al., 2012). Relevant research indicates that global warming

and precipitation reduction have gradually intensified the disastrous effects of dry-hot wind, with the

frequency of the regional occurrence of these effects having gradually increased (Liu et al., 2012).

Chen et al. (2001) found that while the occurrence of dry-hot wind in wheat production gradually

decreased from the 1960s to the 1990s, the occurrence of dry-hot wind has increased in recent decades.



This increase has brought about severe harms during the grouting stage of winter wheat (Zhao et al.,
2012). Liu et al. (2008) reported that within the backdrop of global warming and global drought, the
occurrence of winter wheat dry-hot wind in Gansu province has gradually intensified in frequency.
This increased frequency has caused relatively large damages to winter wheat, and it also indicates
that the occurrence of dry-hot wind is sensitive to global climatic changes. Wang et al. (2010)
conducted research on the forecast and prediction of dry-hot wind disasters. Some scholars
implemented comprehensive research on the occurrence intensity of two kinds of dry-hot wind: light
and heavy dry-hot winds. These scholars established indexes for evaluating occurrences and
constructed models for the evaluation of loss in disasters according to the extent of disasters afflicting
winter wheat (Piao et al., 2010; Pan et al., 2011; Wu et al., 2012; Lobell et al., 2011). Therefore, under
the auspices of climatic changes, scholars have carried out numerous research studies on such aspects
as the rules of disaster occurrence in agricultural ecosystems and adaptations to climatic changes.
However, due to the characteristics of the regional occurrence of agricultural meteorological disasters,
the same disaster may vary between different regions. In recent years, the frequency of occurrence of
meteorological disasters has intensified with the rapid development of China's agricultural ecosystems.
Winter wheat production in the typical agricultural areas of Huang-huai-hai region will experience
severe challenges because existing research knowledge does not capture the situation of climatic
changes in different regions. Therefore, regional implementation of research on the rules governing
meteorological disasters occurring amongst the backdrop of climatic changes is helpful for the safe
production of winter wheat in the research areas. This research also provides a theoretical basis for the
prevention and alleviation of agricultural meteorological disasters.
As a main economic area in the Central-plains Economic Zone, Henan province has an important




strategic position and is an important base for the production of China's vital agricultural crops.
Located in the typical agricultural area of the Huang-huai-hai plain, the city of Shangqiu represents a
strategically pivotal city in the Central-plains Economic Zone. Due to its unique geographical position,
Shangqiu is a major region for agricultural development in Henan province and represents an
important base for the production of marketable grain and subsidiary agricultural products in China.
In Shangqiu, winter wheat experiences damages caused by cold and frost, as well as destruction
caused by drought and dry-hot wind throughout the growth period of wheat. Among these destructive
forces, dry-hot wind represents one of the major agricultural calamities that bring about severe
damages to winter wheat during its late growth period. The analysis in this paper uses the
daily-recorded meteorological data from 1963 to 2012 in combination with the index system of winter
dry-hot wind to quantitatively analyze the features, effects, and changing trends of dry-hot wind on
winter wheat production in Shangqiu within the past 5 decades. In the face of climatic changes, this
study provides a policy-making basis for safe production of agricultural products, damage avoidance,
and disaster prevention and alleviation.
**2. Materials and methods**
**2.1. Site description**
Located at 32°00′ N-40°30′ N, 113°00′E-121°00′E, the Huang-huai-hai plain has an area of
approximately 38.7 km$^2$. It contains the waters from the Huanghe (the Yellow River), the Huaihe, and
the Haihe. The Huang-huai-hai plain represents an important base for agricultural production in China;
the main crops in this plain are winter wheat, corn and cotton. This plain has a soil type of zonal
brown or cinnamon, a generally warm temperate humid or semi humid climate, an annual temperature
within 14-15℃, a precipitation of 500-1000 mm (with large variations annually), and an annual



temperature accumulation above 0℃ of 4500-5500℃. Located in the eastern area of Henan province,
Shangqiu is a typical agricultural area in the Huang-huai-hai plain. Located between 114°49′
E-116°39′ E, 33°43′ N-34°52′ N, Shangqiu has a total area of 10704 km$^2$. The soil in the research area
is moist, and the climate is a typical warm and semi-humid continental monsoon climate with an
average annual temperature within 13.9℃-14.3℃, an average annual precipitation of 623 mm, an
average annual sunshine duration of 2204.4-2427.6 hours, and an average frost-free period of 207-214
days. The test area is characterized by a warm and windy climate in the spring, warming and
concentrated rainfalls in the summer, cooling and long-term sunshine in the autumn, and cold,
low-snow conditions in the winter.
**2.2 Methods and data Collection**
In Shangqiu, winter wheat enters the flowering stage from late April to early May, while winter wheat
in the western and northern areas of Henan province enters this stage about a week later than winter
wheat in other areas (Cheng et al., 2011). In the majority of Henan province, winter wheat enters the
grouting stage in the middle of May. Therefore, in this paper, daily meteorological data of every year
was selected from late May to early June (from May 21 to June 5) The effects of dry-hot winds on
winter wheat from the grouting stage to the maturation stage (the late growth stage) were
systematically analyzed.
Daily meteorological data recorded from 1963 to 2012 at eight agricultural meteorological
observatories in Shangqiu City, Minquan, and Suixian were selected, and 3 meteorological factors
(daily maximum temperature, relative humidity at 2:00 pm, and wind speed at 2:00 pm) were adopted
as the basis for analysis. Meanwhile, Mann-Kendall mutation tests were applied to the analysis of the
timing rules of the meteorological factors of dry-hot wind disasters (Zhou et al., 2000; Wei, 2007).





Sample distribution of this method does not necessarily follow certain rules and is immune to the
perturbations from individual abnormal values in the sample. The mutation points were systematically
analyzed by the form and the directional trend of the cumulative departure curve to identify the
genuineness of these points. The mutation points were calculated by utilizing programs such as Origin
8.5, Mann-Kendall mutation tests, and Excel. The meteorological data were provided by the
meteorological bureau and agricultural bureau of Shangqiu City. All the data were acquired from
observations according to the requirements of the *Agricultural Meteorological Observation Standard*
issued by the China Meteorological Bureau, and the methods for observations remained uniform.
**2.2 Selection of dry-hot wind indexes**
In this paper, dry-hot winds featuring high temperatures and low humidity were mainly analyzed.
Such dry-hot winds are the main type of wind that brings about damages to winter wheat in Shangqiu
at the late growth period and generally occur at relatively high frequencies in middle and late May and
early June (Zhao et al., 2012). The concrete indexes for analysis are referred to in *Disaster Grades of*
*Dry-hot Wind in Wheat* (Huo et al., 2007) (Table 1).

**Table 1.** Disaster grades of dry-hot wind

| Light | | | Heavy | | |
|---|---|---|---|---|---|
| daily maximum temperature (℃) | relative humidity at 2: 00 pm (%) | wind speed at 2: 00 pm (m/s) | daily maximum temperature (℃) | relative humidity at 2: 00 pm (%) | wind speed at 2: 00 pm (m/s) |
| ≥32 | ≤30 | ≥3 | ≥35 | ≤25 | ≥3 |

**3. Results and analysis**
**3.1 Changes of the meteorological factors of dry-hot wind**
The flowering and grouting stages of winter wheat from 1963 to 2012 in Shangqiu were quantitatively
analyzed by applying the methods of Mann-Kendall mutation tests. Three meteorological factors were
analyzed: daily maximum temperature, relative humidity at 2:00 pm, and wind speed at 2:00 pm at a



height of 10 m. The threshold line was set as: $Y=\pm1.96$ (at the level of $p$=0.05); results are listed in Fig.
2. It shown that the fluctuations in the curve depicting the ordinal and inverse sequence statistics of
these 3 factors were relatively large, indicating relatively significant rates of annual changes (Figure
2). The ordinal sequence statistics of daily maximum temperature and wind speed at 2:00 pm were
essentially below 0, showing that within the recent 5 decades, daily maximum temperature and wind
speed at 2:00 pm exhibited a significant declining trend. The ordinal sequence statistics of relative
humidity at 2:00 pm were mostly above 0 and exhibited an increasing trend in fluctuations, indicating
that within the recent 5 decades, relative humidity at 2:00 pm exhibited a significant increasing trend.
In this research, by analyzing ordinal sequence curves (UF curves) and inverse sequence curves (UB
curves), in combination with cumulative departure curves, the genuine mutation points of every factor
were analyzed and evaluated. It shown that the intersection point of the ordinal and inverse sequence
curves of the maximum temperature appear in 1972, after which the UF curve does not exceed the
critical value line (Figure 1a). From 1972 to 1982, the cumulative departure curve (Figure 1b)
corresponding to the ordinal and inverse sequence curves of the maximum temperature exhibit first an
increasing trend and then a decreasing trend. Therefore, around 1972, a mutation gradually increased
in appearance in the maximum temperature at the late growth stage of winter wheat. However, this
observation does not reach a level of significance ($p$>0.05). Sequence curves of relative humidity at
2:00 pm intersect in the years 1968, 1981, and 1984, and the major parts of the UF and UB do not
exceed the critical value line (Figure 1c). The minimum value of relative humidity at 2:00 pm occurs
on the cumulative departure curve in 1981 (Figure 1d). After 1981, this value exhibits an increasing
trend. Therefore, around 1981, a significant ($p$<0.05) mutation gradually increases in appearance in
the value of relative humidity at 2:00 pm. No intersection point appears in the ordinal and inverse




sequence curves (UF and UB curves) of wind speed at 2:00 pm, and both curves exceed the critical
value line (Figure 1e). Therefore, the mutation tests failed. However, the wind speed at 2:00 pm
reaches its peak value in 1984 (Figure 1f) and then exhibits a trend of gradual decline. Therefore,
around 1984, a significant ($p<0.05$) mutation of gradual decline appears in the value of wind speed at
2:00 pm.

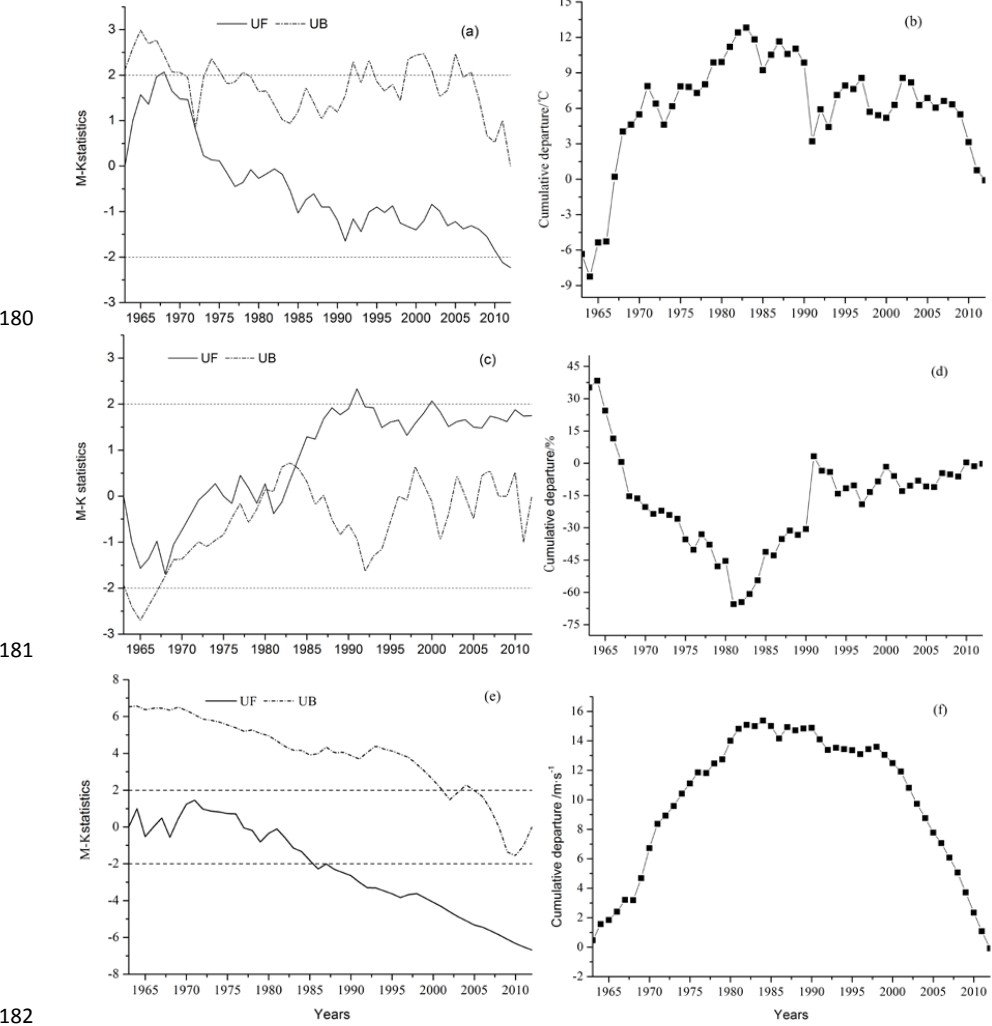




**Figure 1.** Mutation test and cumulative departure of daily maximum temperature, relative humidity at 2:00 pm and

wind speed at 2:00 pm.

Note: daily maximum temperature (a, b), relative humidity at 2:00 pm (c, d) and wind speed at 2:00 pm (e, f)





**3.2 Days of dry-hot wind occurrence**
**3.2.1 Light dry-hot wind**
From 1963 to 2012, the average number of days of the occurrence of high-temperature and low
humidity light dry-hot wind in winter wheat exhibited a general trend of fluctuating decline (Figure 2).
The average number of days for light dry-hot wind occurrence fluctuated within 0-5.9 days, with an
average value of 1.5 days, a variation coefficient (CV) of 83.3%, and a standard error of 1.3 days.
Over the past 50 years, relatively severe occurrences of light dry-hot wind appeared in 1965 and 1981,
and relatively less severe occurrences of light dry-hot wind appeared in 1993, 1996, 2007 and 2012.
No occurrence of light dry-hot wind appeared in 1963, 1984, 1985, 1991 and 2010, and the maximum
number of days for light dry-hot wind occurrence appeared in 1965, totaling 5.8 days. From the fitted
equations, it can be concluded that from 1963 to 1980, the number of days for light dry-hot wind
occurrence basically stabilized at a certain level. From 1981 to 1996, the number of days of light
dry-hot wind decreased rapidly, while from 1997 to 2012, the number of days decreased more slowly.
This decrease was correlated to the comprehensive effects of temperature, water, and wind speed after
the growth period of winter wheat.

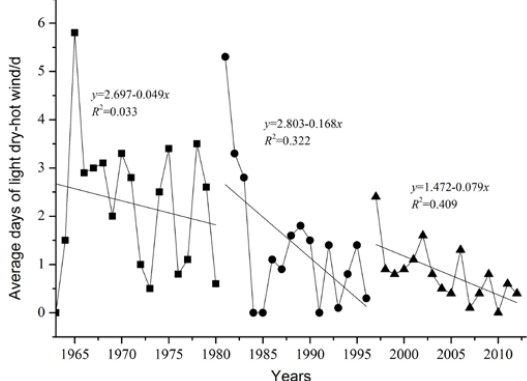


**Figure 2.** Changes in annual average days of light dry-hot wind



By analyzing the characteristics of the annual changes of light dry-hot wind, it can be observed that in
the 1960s, Shangqiu witnessed the severest occurrence of light dry-hot wind, with a total of 2.6 days.
This occurrence was followed by occurrences of light dry-hot wind in the 1970s, 1980s and 1990s,
with the number of days reaching 2.1 days, 1.7 days, and 0.9 days, respectively. The past 10 years
have experienced the lightest damages caused by light dry-hot wind, with the number of days totaling
0.8 days.
**3.2.2 Heavy dry-hot wind**
From 1963 to 2012, the average number of days of the occurrence of heavy dry-hot wind in winter
wheat exhibited a general trend of fluctuating decline (Figure 3), with the average number of days
amounting to 0.5 days. Through calculation, it can be concluded that the variation coefficient (CV)
amounted to 98.9% with a standard error of 0.8 days. Over the past 50 years, relatively severe
occurrences of heavy dry-hot wind appeared in 1967, 1968, and 1994, peaking in 1968 at 3.5 days.
Relatively less severe occurrences of heavy dry-hot wind appeared in 1977, 1987, and 1999. From the
fitted equations, it can be concluded that from 1963 to 1981 and from 1997 to 2012, heavy dry-hot
wind occurrence decreased slowly, while from 1982 to 1996, occurrences increased slowly.

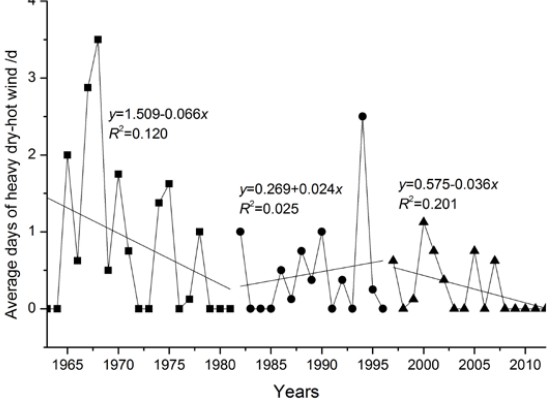


**Figure 3.** Changes in annual average days of severe dry-hot wind



By analyzing the annual changes of heavy dry-hot wind, it can be observed that in the 1960s,
Shangqiu witnessed the severest occurrence of light dry-hot wind in the late growth period of winter
wheat with an average number of days of 1.4 days. This severity is followed by the occurrences of
heavy dry-hot wind in the 1970s, 1990s, and the past 10 years, with the annual number of days for
occurrence reaching 0.7 days, 0.5 days, and 0.4 days, respectively. In the 1990s, Shangqiu witnessed
the smallest damages caused by heavy dry-hot wind occurrence, with the number of days totaling 0.3
days.
**3.3. Effects of climatic changes on meteorological disasters caused by dry-hot wind**
The correlations between the number of days of dry-hot wind occurrence and climatic factors are
shown in Table 2. It indicated the number of days of dry-hot wind occurrence was highly correlated
with 10 meteorological factors, with coefficients ranging from -0.639 to 0.753. Apart from the
significance levels ($p$=0.05) of the correlation coefficients between the number of days and the
average maximum temperature, between the number of days and the average precipitation, and
between the number of days and the average evaporation, the correlation coefficients between the
number of days of dry-hot wind occurrence and the remaining factors all indicated high levels of
significance ($p$<0.01).
**Table 2** Correlations between day number of dry-hot wind occurrence and climatic factors

| | Average temperature | Average maximum temperature | Average minimum temperature | Day number of maximum temperature≥30℃ | Day number of maximum temperature≥32℃ |
|---|---|---|---|---|---|
| Correlation Coefficient | 0.612 | 0.498 | 0.414 | 0.701 | 0.753 |
| Sig. | 0.01 | 0.01 | 0.04 | 0.01 | 0.01 |
| | Day number of maximum temperature≥35℃ | Average relative humidity | Average precipitation | Average evaporation | Average day number of precipitation |
| Correlation Coefficient | 0.594 | -0.604 | -0.497 | 0.408 | -0.639 |
| Sig. | 0.01 | 0.01 | 0.03 | 0.05 | 0.01 |





The occurrence of dry-hot wind disasters exhibits the sensitive response to global climatic changes.
With the influences of climatic warming causing decreased relative humidity, reduced number of days
of precipitation, decreased precipitation amount, increased average temperature, enhanced average
maximum and minimum temperatures, and gradually increasing average evaporation, dry-hot wind
disasters occur with relatively stronger intensities, higher frequencies, and more severe damages. On
the contrary, during the period of moderate and cooling weather, dry-hot wind disasters occur less
frequently with weak intensities.
**3.4 Days of dry-hot wind occurrence and winter wheat yield**
The correlation between winter wheat yield per unit area and the number of days of dry-hot wind
occurrences in the past 20 years (from 1991 to 2012) is shown in Figure 4. It shown that the number
of days of dry-hot wind occurrences exhibits a fluctuating trend of decline, whereas winter wheat
yield per unit area exhibits a trend of enhanced fluctuations, indicating that the greater the number of
days of dry-hot wind occurrences, the lower the winter wheat yield per unit area. Namely, these
variables were significantly negatively correlated ($p<0.05$). In Shangqiu, during the late growth period
of winter wheat, the trend of average maximum temperature change was different from that of average
temperature change, and no significant rise was observed (Figure 1b). This pattern of temperature
change was relatively beneficial for winter wheat at the grouting stage. Relative humidity at 2:00 pm
exhibits an increasing trend, albeit slowly. Wind speed exhibits a trend of decline (Figure 1d and 1f).





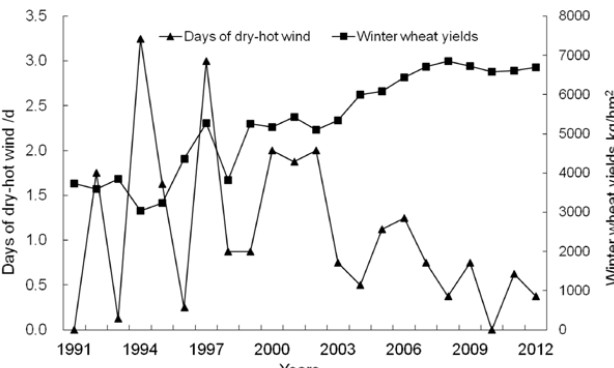


**Figure 4.** The relationship between dry-hot wind days and yield of winter wheat

**4. Discussion**
In Shangqiu, the range and frequency of dry-hot wind occurrences during the growth period of winter
wheat generally exhibited a trend of gradual decrease, and the corresponding occurrence frequency of
the disasters counted by the sliding curves also showed a trend of gradual decline. In 1972, significant
($p<0.05$) mutations gradually increase in appearance in the maximum daily temperature related to
dry-hot wind disasters. Around 1984, significant decreases appeared in wind speed at 2:00 pm,
whereas relative humidity at 2:00 pm increased markedly. Around 1981, the relative humidity value
experienced a conspicuous gradual increase. The magnitude for temperature increase in China
amounted to 0.22℃/10 years, reaching 0.25℃/10 years from 1963 to 2012 (Tan et al., 2009; Zhu et al.,
2012a). The average temperature in Shangqiu increased commensurately (Shi et al., 2012). However,
the trend of maximum daily temperature change was different at different stages of the winter wheat
growth period (Tan et al., 2009; Xiong et al., 2010). In Shangqiu, at the late growth period of winter
wheat, the maximum daily temperature did not significantly increase with average temperature (Zhu
et al., 2012b), which was beneficial for winter wheat at the grouting stage. Under global climatic
changes, the decreased relative humidity of air is the main reason for drought (Jin et al., 2009). In this
research, from 1963 to 2012, the relative humidity at 2:00 pm increased slowly, which was relatively





in accordance with the research results of Zhao et al.(2012) and Cheng et al. (2011).
Over the past 50 years, the overall disasters of dry-hot wind in winter wheat exhibited a gradual
decreasing trend. Regional and periodic disasters of dry-hot wind still exist because of differences in
matching the meteorological factors of temperature, water, and wind speed in different regions and at
different times. Therefore, to minimize the harmful effects of dry-hot wind on winter wheat at the late
growth period during the process of agricultural production, emphasis should be placed on the
prevention of dry-hot wind disasters, and research concerning aspects other than climatic factors
should be intensified (Sridhar et al., 2006). In recent years, the existing indexes of dry-hot wind and
concomitant research results cannot meet the requirements of regional food production and the
prevention of agricultural meteorological disasters. Relatively huge differences exist in climatic
environments, soil, and crop types of different regions in China (Zhao et al., 2012). In addition, as
differences also exist in the mechanisms for the effects of dry-hot wind on different food crops, new
generations of experimental research concerning the indexes of dry-hot wind should be continuously
implemented. Meanwhile, under the auspices of global climatic changes, the harmful effects of
dry-hot wind disasters was correlated with the physiological structural features of agricultural crops,
developmental processes, and degrees of regional environmental effects. Therefore, the influences of
human activity, different policies on field management, and the resistances of winter wheat with
different qualities should also be taken into consideration (Jung et al., 2010;Zhu et al., 2012 ).
Dry-hot wind generally occurs at the late growth period of winter wheat and poses relatively severe
threats to winter wheat at the grouting stage (Chen et al., 2001; Zhao et al., 2012). When it occurs,
dry-hot wind exerts relatively huge effects on the yield, 1000-seed weight, and quality of winter wheat
(Benzian et al., 1986; Li et al., 2003). These impacts are in accordance with the results of research that





indicate that in Shangqiu, the average annual number of days of dry-hot wind occurrence was
significantly negatively correlated with winter wheat yield per unit area. However, numerous factors
influence the yield of winter wheat, including biological technologies, investment in agricultural
production (including agricultural chemicals and fertilizers), and other meteorological factors. In this
research, only the effects of dry-hot wind on winter wheat yield were analyzed. In the future, such
effects should be comprehensively analyzed in combination with other factors, including biological
gene technologies, crop cultivars, and crop diseases and pests.
**5. Conclusions**
The range and frequency of dry-hot wind exhibited tended to decline in the recent 50 years. The
significant inter-annual variations were related with the corresponding meteorological conditions of
temperature, moisture and wind speed. The most serious damages of light and severe dry-hot wind
both occurred in 1960s while the damages appeared less in the 1980s and the last decade, which could
be also explained by the corresponding temperature, moisture and wind speed conditions. The
comprehensive effects of daily maximum temperature, relative humidity at 2:00 pm, and wind speed
at 2:00 pm showed that in Shangqiu, disasters of dry-hot wind in winter wheat generally exhibited a
general trend of gradual decline, and a remarkably decreased wind speed played the main role in
mitigating the overall disasters of dry-hot wind. Annual average days of dry-hot wind had a great
influence on the yields of winter wheat in Shangqiu.
**Acknowledgements**
The study was supported by the Natural Science Foundation of China (41140019; 41501263); Programs of Science and
Technology of Henan Province, NO. 132102310357 and NO. 142102310299. The authors wish to thank anonymous
reviewers for their useful comments and suggestions on the manuscript.





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
