# Peer review of "Analysis of how dry-hot wind hazard has changed for winter"

_Natural Hazards and Earth System Sciences, 2015_

## Referee Comment (RC1) · Anonymous Referee #1 · 14 Mar 2016

Review of the manuscript entitled:

"Analysis of how dry-hot wind hazard has changed for winter wheat in the Huang-huai-hai plain"

By Benlin Shi, Xinyu Zhu, Hongzhong Li, Yunchuan Hu and Yi Zhang

Manuscript Number: nhess-2015-330

GENERAL COMMENTS

This manuscript deals with the climatic conditions during the last fifty years over central east China that are considered of relevance for the wheat harvest in winter time in this region. The document presents an evaluation of the temporal evolution of some climatological variables and their connection to the variations of wheat production. Therefore, this represents an impact case study about the potential implications of the climate evolution in the agricultural sector. Studies like the present one are of great value for the impact community and its interest relies on the their ability to detect causalities between the climatic forcing and the effect on societal areas of primary importance. However, the work presents a bit too shallow analysis of the results and the methodological aspects are somehow too superficially elucidated.

Based on the above concerns I suggest a major revision of the manuscript. Some suggestions and comments follow to help disentangling the main issues addressed above. The authors should address them prior a potential publication of the paper.

SPECIFIC COMMENTS

a) Major Comments (MC)

MC1. As commented above, one of the major issues refers to how the methodological aspects are presented in the manuscript. In my opinion a too technical language within a superficial description of the premises and outcomes of methodology obscure the interpretation of the results and it is somehow unbalanced. This would difficult the understanding of a potential reader about the author's reasoning on the outcomes of the manuscript.

First of all, the abstract includes too detailed results. It should instead include an overview of the main ideas of the work considering the focus and the methodologies applied in a shallow fashion and definitely a general statement about the achievements and conclusions of the paper. I would for instance consider eliminating the lines 32 to 39, where it goes too much into the detail of single years response to the analysis.

MC2. The ideas within the Introduction are a bit muddled, some order in the sequence of ideas will benefit the comprehension of the motivation of this work: first of all, it is necessary an explanation about the concept of "dry-hot wind" as many potential readers are not necessarily familiar with such terminology; second the authors might

explain why is this topic relevant in the region, highlighting what are the aspects of the climate variability in the region with a potential impact on the winter harvest; finally it should be stated how the authors will address the questions posed. All in all, I also find a general tendency along the text to repeat unnecessarily some ideas, e.g., the paragraph at the beginning of Section 3.1 is overstated as it was already explained before in Section 2.2 (where it is mentioned which climatological variables are used for the present work).

MC3. "The Methods and data Collection" (please, mind when to use capitol letters) is poorly described. Apart from some details that I will outline in the section below, the authors should provide some hints why the Mann-Kendall mutation test was selected in their study (further than citing a couple of previous works).

MC4. An explanation about the methodology is strongly recommended. What are the expectations from the use of such methodologies? What is the basis on which the method is useful to understand the connections between the climatological and agricultural records?

MC5. It is of great importance to guide the reader across the manuscript; therefore the authors should provide a more clear explanation about: the meaning of mutation points (Line 136, in addition there is no need to specify that the programming was done in Excel or Origin, at least it is clear that it contributes less to the overall understanding of the paper); the definition of dry-hot wind (Line 142) arrives, as said before, a little too late in Section 2.2; the reader could understand that the threshold line is equivalent 1 sigma, but it should be clarified directly, what are the implications (in Line 153) when a series exceeds the named threshold line (it can be extrapolated from the text and from the knowledge of the reader, but in my view the paper would largely benefit from a more clarifying reading), the same applies to the concepts of ordinal and inverse sequence statistics (Line 154), what do we learn from these curves and from the cumulative departure curves (Line 162)?

[Figure]

MC6. As a result of the scant explanations about methodological aspects and terminology, the reader might wonder what do we learn from the sequences plotted in Figure 1 that cannot be explored in an initial stage just from the anomalies of the temperature, humidity and wind? Therefore, it can be said that to some extent, Figure 1 is poorly explained across the text. Is the methodology based on raw series or in anomalies? In the case of the latter, with respect to what reference period?

MC7. Sections 3.3, 3.4 and the Discussion are in general a bit discouraging as they do not offer a comprehensive view about the claims that the manuscript proposed at the beginning in the abstract or in the Introduction: there is an overall over-use of the concept climate change, specially when apparently there is in general some disconnection between the increase of temperatures over the region as a consequence of the global warming trends while the amount of days categorized as dry-hot wind days decreases and the winter wheat production has broadly increased. Section 3.3 just provides results that arise more from a logical thinking about climatic variations during the last decades than from what data and graphs shows in this manuscript (e.g., paragraph between Lines 237-243). The relevance of this manuscript lies on the exercise of exploring the climatic and harvesting data and their interconnections. It is a reality that adaption should be a primary target of societies, but this paper just stresses too much this idea while it sacrifices a straight approach showing how to relate the different type of data. The authors could make some effort in disentangling the too repetitive discourse and show plain but still valid approach and results.

b) Further comments (FC)

FC1. The keyword grain filling does not appear in the whole manuscript. Would the authors revise the role of the selected keywords?

FC2. I would recommend to revisit the title of the manuscript stating it more clearly that an analysis of the climatological conditions over the region of interest is presented and their connection to the wheat production is evaluated (". . . how the dry-hot wind hazard

has changed for the winter wheat. . ." is a bit misleading).

FC3. In Line 159 it is stated that the ordinal sequence "exhibited an increasing trend in fluctuations, indicating that within the recent 5 decades, relative humidity at 2:00 exhibited a significant increasing trend". This is an example connected to the comment in MC6 in this manuscript: what is the difference between both trends? Could the authors clarify what do we learn from this ordinal sequence increase compared to what we would learn from the raw trends of the variables? Certainly the text will highly benefit from clarifications in this sense to help interpret results.

FC4. What does it imply that a mutation gradually increased in appearance in Line 167? And where or how is the statistical significance evaluated?

FC5. Could the authors explain why the mutation tests failed in Line 176?

FC6. Could the authors illustrate more rigorously the meaning of the fitted equations in Line 195? Do they refer to linear regressions over the series of the dry-hoy days? What do they need this equations for is not yet clear in the documents. Neither it is obvious in the text why the authors adjust the data in three different regressions for different periods (1963-1980, 1981-1996 and 1997-2012).

FC7. It would be elementary to gain some insight into the differences between the impact of the light and the heavy dry-hot days on the winter harvest, provided that there is such classification in Section 3.2 of the manuscript.

TECHNICAL COMMENTS (TC)

TC1. Line 11. Should read ". . .a key constraint on agriculture".

TC2. Line 23. "We synthesized analyzed the distribution. . ."? Is there some extra word in this sentence?

TC3. Revise the expression in Line 30 (". . . while damages appear less in the...").

TC4. Line 42: ". . .through improving filed microclimatic condition"?

[Figure]

TC5. References on Line 47? (they do appear a bit too late in Line 50).

TC6. Cite dates in Line 60.

TC7. Instead of "Some scholars", cite some refs in Line 72.

TC8. Review the meaning of the sentence "the frequency of occurrence of meteorological disasters has intensified with the rapid development of China's agricultural ecosystems" in Lines 80-81. Is the sense correct?

TC9. Define "temperature accumulation" in Line 111.

TC10. Would a map of the area under study be of great visual help in Section 2.1?

TC11. Add a brief explanation about why the meteorological records are taken at 2:00 pm. Is that the single sampling time per day? Once it is mentioned for the first time, the authors do not need to state that the variables are measured at 2:00 pm every time along the manuscript.

TC12. What does it mean, "Sample distribution of this method does not necessary follow certain rules..." in Line 133?

TC13. In Line 154 it should read, "It is shown...".

TC14. Caption of Figure 1: comprise the two sentences in only one including the references to the panels (a - f).

TC15. Is the average of the number of light dry-hot days what fluctuates between 0 and 5.9? Or is it just the number of days, where the average is then 1.5 days (Lines 190-191). Afterwards it is said that the maximum number of light dry-hot days is 5.8. Review the numbers here please.

TC16. Could the authors find a more compact way of expressing their results than citing sequentially all years with no occurrence of dry-hot days (e.g., Line 194)?

TC17. Line 229, it should read, "It indicated that..."

TC18. What do the authors refer to with. . .", during the period of moderate and cooling weather" in Line 242?

TC19. Line 246, it should read "It is shown that the number. . ."

TC20. What do the "sliding curves" mean in Line 260?

TC21. Line 261, ". . . mutations gradually increased. . ."

TC22. The paper will largely benefit from a native English speaker reviews.

I encourage the authors to accomplish these comments and suggestions for a better and easier understanding of their work in order to end up with a comprehensive piece of work that could be then published.

---

## Referee Comment (RC2) · Anonymous Referee #2 · 25 Mar 2016

The paper demonstrates the the effects of climate change on wheat production. Shangqiu is selected as the case study area to determine the occurrence regularity of dry-hot wind hazard on winter wheat production in Huang-huai-hai plain. The topic is useful for wheat production.

However, there are some questions as follows.I suggest major revision. 1. In the paper, the insignificant results are not reliable and helpful, such as in the introduction: From 1963 to 2012, a climatic mutation point of daily maximum temperature was found near 1972, but insignificantly (p>0.05).

2. The second paragraph in the introduction should be modified. The first two sencentes in this paraphagh should be in the form of scientific paper, not like news. And the content in this paraphagh should be reorganized, maybe divided into two parts.

[Figure]

3.Besides, the English should be improved for better understanding

---

## Author Comment (AC1) · 3 May 2016

GENERAL COMMENTS

This manuscript deals with the climatic conditions during the last fifty years over central east China that are considered of relevance for the wheat harvest in winter time in this region. The document presents an evaluation of the temporal evolution of some climatologically variables and their connection to the variations of wheat production. Therefore, this represents an impact case study about the potential implications of the climate evolution in the agricultural sector. Studies like the present one are of great value for the impact community and its interest relies on the their ability to detect causalities between the climatic forcing and the effect on societal areas of primary

importance. However, the work presents a bit too shallow analysis of the results and the methodological aspects are somehow too superficially elucidated.

Based on the above concerns I suggest a major revision of the manuscript. Some suggestions and comments follow to help disentangling the main issues addressed above. The authors should address them prior a potential publication of the paper.

R: Thank you for the comments. We changed the title of our manuscript as suggested and revised some of the Abstract (Line 29-37). We have also invited a native English speaker and a professional editing company to edit our manuscript. We added a new figure (Figure 1) to describe the study areas. Some data analysis methods were added (Line 124-145). Section 3.1 was re-written, and some of results were revised (Line 155-157). Moreover, we re-analyzed Sections 3.3 and 3.4 and added Figure 5 to replace the original Figure 4 to show the influences of light and heavy dry-hot winds on winter wheat yield. The revised contents are as follows: Line 192-195, Line 208-210, Line 217-218, Line 225-234 and Line 237-244.

a) Major Comments (MC)

MC1. As commented above, one of the major issues refers to how the methodological aspects are presented in the manuscript. In my opinion a too technical language within a superficial description of the premises and outcomes of methodology obscure the interpretation of the results and it is somehow unbalanced. This would difficult the understanding of a potential reader about the author's reasoning on the outcomes of the manuscript.

First of all, the abstract includes too detailed results. It should instead include an overview of the main ideas of the work considering the focus and the methodologies applied in a shallow fashion and definitely a general statement about the achievements and conclusions of the paper. I would for instance consider eliminating the lines 32 to 39, where it goes too much into the detail of single years response to the analysis.

R: Thank you for the comments. In the revised manuscript, we rewrote some of the Abstract and added some important conclusions. The revised text can be found on Line 29-37.

MC2. The ideas within the Introduction are a bit muddled, some order in the sequence of ideas will benefit the comprehension of the motivation of this work: first of all, it is necessary an explanation about the concept of "dry-hot wind" as many potential readers are not necessarily familiar with such terminology; second the authors might explain why is this topic relevant in the region, highlighting what are the aspects of the climate variability in the region with a potential impact on the winter harvest; finally it should be stated how the authors will address the questions posed. All in all, I also find a general tendency along the text to repeat unnecessarily some ideas, e.g., the paragraph at the beginning of Section 3.1 is overstated as it was already explained before in Section 2.2 (where it is mentioned which climatologically variables are used for the present work).

R: Thank you for your constructive suggestions. As suggested, we rewrote some of the Introduction, and the revised text can be found on Line 45-51, Line 52-55, Line 60-67, Line 75-79 and Line 83-87. Regarding the beginning of Section 3.1, we agree with your comments and deleted the duplicated sentences.

MC3. "The Methods and data Collection" (please, mind when to use capitol letters) is poorly described. Apart from some details that I will outline in the section below, the authors should provide some hints why the Mann-Kendall mutation test was selected in their study (further than citing a couple of previous works).

R: Thanks. We changed "The Methods and data Collection" to "Data collection and methods ". Moreover, we added some additional content about the Mann-Kendall mutation test. The revised contents can be found on Line 132-143.

MC4. An explanation about the methodology is strongly recommended. What are the expectations from the use of such methodologies? What is the basis on which

the method is useful to understand the connections between the climatologically and agricultural records?

R: Thank you for the comments. In the revised manuscript, we have provided an explanation of the Mann-Kendall mutation test (Line 132-139). Line 135-138 describe the requirements for the use of this method. We also provided some additional text to explain the connections between the climatological and agricultural records (Line 132-135, Line 141-143).

MC5. It is of great importance to guide the reader across the manuscript; therefore the authors should provide a more clear explanation about: the meaning of mutation points (Line 136, in addition there is no need to specify that the programming was done in Excel or Origin, at least it is clear that it contributes less to the overall understanding of the paper); the definition of dry-hot wind (Line 142) arrives, as said before, a little too late in Section 2.2; the reader could understand that the threshold line is equivalent 1 sigma, but it should be clarified directly, what are the implications (in Line 153) when a series exceeds the named threshold line (it can be extrapolated from the text and from the knowledge of the reader, but in my view the paper would largely benefit from a more clarifying reading), the same applies to the concepts of ordinal and inverse sequence statistics (Line 154), what do we learn from these curves and from the cumulative departure curves (Line 162)?

R: Thank you for the comments. In the revised manuscript, as suggested, the original text in Line 136 was deleted. The original text in Line 142 was also deleted, and in the Introduction, we defined dry-hot wind (Line 45-51). According to the comments of MC5, we rewrote the Section 3.1, and the revised text can be found on Line 155-177.

MC6. As a result of the scant explanations about methodological aspects and terminology, the reader might wonder what do we learn from the sequences plotted in Figure 1 that cannot be explored in an initial stage just from the anomalies of the temperature, humidity and wind? Therefore, it can be said that to some extent, Figure 1 is poorly

explained across the text. Is the methodology based on raw series or in anomalies? In the case of the latter, with respect to what reference period?

R: Thank you for the comments. We added a relevant introduction of the Mann-Kendall mutation tests in Section 2.2 (Line 132-139). Because we did not originally explain Figure 1 clearly, based on your comments, we have re-written Section 3.1 and improved the explanation of Figure 1 (Line 155-177). We hope that this revision will meet with your approval.

MC7. Sections 3.3, 3.4 and the Discussion are in general a bit discouraging as they do not offer a comprehensive view about the claims that the manuscript proposed at the beginning in the abstract or in the Introduction: there is an overall over-use of the concept climate change, specially when apparently there is in general some disconnection between the increase of temperatures over the region as a consequence of the global warming trends while the amount of days categorized as dry-hot wind days decreases and the winter wheat production has broadly increased. Section 3.3 just provides results that arise more from a logical thinking about climatic variations during the last decades than from what data and graphs shows in this manuscript (e.g., paragraph between Lines 237-243). The relevance of this manuscript lies on the exercise of exploring the climatic and harvesting data and their interconnections. It is a reality that adaption should be a primary target of societies, but this paper just stresses too much this idea while it sacrifices a straight approach showing how to relate the different type of data. The authors could make some effort in disentangling the too repetitive discourse and show plain but still valid approach and results.

R: Thank you for the comments. In the revised manuscript, we re-analyzed the relationships between climatic factors and the days of dry-hot wind occurrence again, and produced a new Table 2. In this new Table 2, we added some content to identify the factors with the greatest impact on the days of dry-hot wind occurrence (Line 225-234). The original Lines 237-243 were revised, and the changes can be reviewed in Lines 228-234. Moreover, according to your suggestions, we had created a new figure (Fig-

ure 5) to replace the original Figure 4 to more clearly demonstrated the differences between the impact of the days of light and heavy dry-hot winds on the winter wheat yields (Line 238-244). Additionally, some parts of the Discussion were revised (Line 249-253, Line 263-267 and Line 277-280).

b) Further comments (FC)

FC1. The keyword grain filling does not appear in the whole manuscript. Would the authors revise the role of the selected keywords?

R: Thank you for your constructive suggestions. We changed the keyword "grain filling stage" to "winter wheat yield".

FC2. I would recommend to revisit the title of the manuscript stating it more clearly that an analysis of the climatologically conditions over the region of interest is presented and their connection to the wheat production is evaluated ("... how the dry-hot wind hazard has changed for the winter wheat..." is a bit misleading).

R: Thank you for your suggestions. We revised the title of our manuscript to "Characteristics of dry-hot wind in the Huang-huai-hai plain and its effect on winter wheat".

FC3. In Line 159 it is stated that the ordinal sequence "exhibited an increasing trend in fluctuations, indicating that within the recent 5 decades, relative humidity at 2:00 exhibited a significant increasing trend". This is an example connected to the comment in MC6 in this manuscript: what is the difference between both trends? Could the authors clarify what do we learn from this ordinal sequence increase compared to what we would learn from the raw trends of the variables? Certainly the text will highly benefit from clarifications in this sense to help interpret results.

R: Thank you for the comments. We have re-written the section 3.1 according to your suggestion. The changed contents were in Line ???-??.

FC4. What does it imply that a mutation gradually increased in appearance in Line 167? And where or how is the statistical significance evaluated?

R: Thank you for the comments. We have re-written the section 3.1, and accordingly this parts (Line 167) were changed to "Therefore, the increasingly abrupt timing of the maximum daily temperature occurred in approximately 1972 during the late growth stage of winter wheat, but this increase did not reach significance (p>0.05) (Figure 2b)." The revised text can be found on Line 155-177.

FC5. Could the authors explain why the mutation tests failed in Line 176?

R: Thank you for the comments. In the new Section 3.1, the reason for the failure is "No intersection point appears in the ordinal and inverse sequence curves (UF and UB curves) of the wind speed at 2:00 pm, and therefore, the mutation test failed (Figure 2e)." (Line 172-174)

FC6. Could the authors illustrate more rigorously the meaning of the fitted equations in Line 195? Do they refer to linear regressions over the series of the dry-hot days? What do they need this equations for is not yet clear in the documents. Neither it is obvious in the text why the authors adjust the data in three different regressions for different periods (1963-1980, 1981-1996 and 1997-2012).

R: Thank you for the comments. First, we added some content to Section 2.2, such as "A mathematical linear fitting equation was used to analyze the variation in the light and heavy dry-hot winds, and according to the variation, if necessary, we could segment a fitting equation to ensure the accuracy of the results." Second, we rewrote Line 195 and explained the reason why we adjusted the data in three different regressions for different periods. The changed contents are in Line 141-143.

FC7. It would be elementary to gain some insight into the differences between the impact of the light and the heavy dry-hot days on the winter harvest, provided that there is such classification in Section 3.2 of the manuscript.

R: Thank you for your comments. In the revised manuscript, according your suggestions, we created a new figure (Figure 5) to replace the original Figure 4 to delineate

the differences between the impact of the light and heavy dry-hot wind days on the winter wheat yields. The changed contents were in Line 238-244.

TECHNICAL COMMENTS (TC)

TC1. Line 11. Should read ". . .a key constraint on agriculture".

R: Thanks. We adopted and changed the sentence.

TC2. Line 23. "We synthesized analyzed the distribution. . ."? Is there some extra word in this sentence?

R: Thanks for your comment. We deleted the word "synthesized" and changed this sentence to "Using regional meteorological data with a daily resolution in the later growth stage of winter wheat from 1963 to 2012, the distribution of annual average days of dry-hot wind in winter wheat growing seasons and the associated responses to the climate change were discussed, as were the relationships between dry-hot wind days and winter wheat yields." Line 20-23.

TC3. Revise the expression in Line 30 (". . .while damages appear less in the...").

R: Thanks. We revised this sentence to "The most serious damage from dry-hot wind occurred in the 1960s, whereas the damage appeared to be less intense in the 1980s and in the last decade, which could also be explained by the corresponding temperature, moisture and wind speed conditions.". (Line 27-29)

TC4. Line 42: ". . .through improving filed microclimatic condition"?

R: We revised this sentence to "Thus, improving the resistance of winter wheat to dry-hot wind hazards should proceed using biological measures, agriculture technology and chemical measures."(Line 36-37)

TC5. References on Line 47? (they do appear a bit too late in Line 50).

R: Thanks for your comment. In Line 41, after the IPCC, we added the references

(IPCC, 2007; 2013).

TC6. Cite dates in Line 60.

R: Thanks. We added the dates in this sentence (Line 55-58).

TC7. Instead of "Some scholars", cite some refs in Line 72.

R: Thanks. We changed "some scholars" to "Liu et al. (2008) and Zhao et al. (2015)....".

TC8. Review the meaning of the sentence "the frequency of occurrence of meteo-rological disasters has intensified with the rapid development of China's agricultural ecosystems" in Lines 80-81. Is the sense correct?

R: Thanks for your comment. This sentence is wrong. We revised it to "In recent years, the occurrence of meteorological disasters increased, and with the rapid development of agriculture in China, winter wheat production in the typical agricultural areas of the Huang-huai-hai region will encounter severe challenges because existing knowledge does not include the effects of climatic changes in different regions." (Line 75-79)

TC9. Define "temperature accumulation" in Line 111.

R: Thanks for your comment. We defined this and changed this to "and an annual accumulation temperature $\geq$ 0°C that varies from 4500-5500°C". Line 104-105.

TC10. Would a map of the area under study be of great visual help in Section 2.1?

R: As per your comments, and we added a new figure ( Figure 1).

TC11. Add a brief explanation about why the meteorological records are taken at 2:00 pm. Is that the single sampling time per day? Once it is mentioned for the first time, the authors do not need to state that the variables are measured at 2:00 pm every time along the manuscript.

R: Thanks for your comment. In the revised manuscript, we rewrote the criteria for

selecting the indicators of the dry-hot wind. (Line 125-132)

TC12. What does it mean, "Sample distribution of this method does not necessary follow certain rules..." in Line 133?

R: In the revised manuscript, this sentence was changed to "The main reason for using Mann-Kendall mutation tests (non-parametric tests) is that compared with parametric statistical tests, non-parametric tests are thought to be more suitable for non-normally distributed data, and have been applied widely to time series of meteorological factors." (Line 135-138)

TC13. In Line 154 it should read, "It is shown. . .".

R: Thanks. We adopt.

TC14. Caption of Figure 1: comprise the two sentences in only one including the references to the panels (a - f).

R: Thanks for your comment. We changed this to "Figure 2. Mann-Kendall mutation test (a, c and e) and cumulative departure of the daily maximum temperature, relative humidity at 2:00 pm and wind speed at 2:00 pm (b, d and f)."

TC15. Is the average of the number of light dry-hot days what fluctuates between 0 and 5.9? Or is it just the number of days, where the average is then 1.5 days (Lines 190-191). Afterwards it is said that the maximum number of light dry-hot days is 5.8. Review the numbers here please.

R: Thanks for your comment. The erroneous numbers (0-5.9) in Line 190 were a clerical error. We changed 5.9 to 5.8 and checked the average (1.5); the average value was correct (reserve one decimal).

TC16. Could the authors find a more compact way of expressing their results than citing sequentially all years with no occurrence of dry-hot days (e.g., Line 194)?

R: Thanks for your comment. To improve the conciseness of the manuscript and address the contextual issue you noted, we deleted this sentence "No occurrence of light dry-hot wind appeared in 1963, 1984, 1985, 1991 and 2010,".

TC17. Line 229, it should read, "It indicated that..."

R: Thanks. We have corrected this issue in the revised manuscript.

TC18. What do the authors refer to with. . .", during the period of moderate and cooling weather" in Line 242?

R: Thanks for your comment. In the revised manuscript, we modified the contents of this paragraph to "The occurrence of dry-hot wind disasters exhibits the sensitive response of weather patterns to global climatic changes. The decreasing trends of the relative humidity and precipitation and the increasing trends of the average temperature, extreme temperatures and average evaporation lead to the occurrence of warming and drying weather. In this weather, dry-hot wind disasters occur with stronger intensities, are more frequent, and cause more severe damage (Table 2). In contrast, during periods of moderate and cooling weather, dry-hot wind disasters occur less frequently and have weak intensities." (Line 228-234)

TC19. Line 246, it should read "It is shown that the number. . ."

R: Thanks for your comment. We have corrected in the revised manuscript.

TC20. What do the "sliding curves" mean in Line 260?

R: Thanks for your comment. In the revised manuscript, we removed the ambiguous words.

TC21. Line 261, ". . . mutations gradually increased. . ."

R: Thanks for your comment. We have corrected in the revised manuscript. The revised text can be found on Line 249-253.

TC22. The paper will largely benefit from a native English speaker reviews.

R: Thanks for your comment. We have invited a native English speaker and a professional editing company to edit our manuscript.

Please also note the supplement to this comment:
http://www.nat-hazards-earth-syst-sci-discuss.net/nhess-2015-330/nhess-2015-330-AC1-supplement.zip

––––––––––––––––––––––––––

---

## Author Comment (AC2) · 3 May 2016

1. In the paper, the insignificant results are not reliable and helpful, such as in the introduction: From1963 to 2012, a climatic mutation point of daily maximum temperature was found near 1972, but insignificantly ($p > 0.05$).

R: Thanks for your comment. This sentence "From1963 to 2012, a climatic mutation point of daily maximum temperature was found near 1972. . . ." was in the Section 3.1. Because we did not explain the Figure 1 clearly, according to your comments, we have re-written the section 3.1 and explained Figure 1 again. The revised text can be found on Line 155-177.

[Figure]

2. The second paragraph in the introduction should be modified. The first two sentences in this paragraph should be in the form of scientific paper, not like news. And the content in this paragraph should be reorganized, maybe divided into two parts.

R: Thank you for your constructive suggestions. In the revised manuscript, the second paragraph in the introduction was modified. The first two sentences in this paragraph were changed to "In recent years, some papers have been published in "Nature", that emphasized the effects of climate change and meteorological disasters on winter wheat, and found that climatic warming and extreme drought resulted in early maturation, yield loss, and decline in dry matter accumulation of wheat (Piao et al., 2010; Lobell et al., 2012; Pongratz et al., 2012;Basso et al., 2014; Asseng et al., 2015)." And we divided this paragraph into two parts. The revised contents in this paragraph were in Line 52-55.

3.Besides, the English should be improved for better understanding.

R: Thanks for your comment. We have invited a native English speaker and a professional editing company to edit our manuscript.

Please also note the supplement to this comment:
http://www.nat-hazards-earth-syst-sci-discuss.net/nhess-2015-330/nhess-2015-330-AC2-supplement.zip
* * *